# Isavuconazole Treatment of Spinal Cord Invasive Aspergillosis Guided by Cerebrospinal Fluid (1,3)-β-d-Glucan Levels in a Patient with Low Interferon-Gamma and Ulcerative Colitis

**DOI:** 10.3390/jof8060557

**Published:** 2022-05-25

**Authors:** Siobhan O’Regan, Brendan O’Kelly, Paul Reidy, Aoibhlinn O’Toole, John Caird, Cora McNally, Samuel McConkey, Eoghan de Barra

**Affiliations:** 1Infectious Diseases Service, Beaumont Hospital, D09 V2N0 Dublin, Ireland; okellybr@tcd.ie (B.O.); paulreidy@mater.ie (P.R.); coramcnally@beaumont.ie (C.M.); smccconkey@rcsi.ie (S.M.); edebarra@rcsi.ie (E.d.B.); 2International Health and Tropical Medicine Department, Royal College of Surgeons Ireland University of Medicine and Health Sciences, D02 YN77 Dublin, Ireland; 3Gastroenterology Service, Beaumont Hospital, D09 V2N0 Dublin, Ireland; aoibhlinnmaryotoole@beaumont.ie; 4Neurosurgery Service, Beaumont Hospital, D09 V2N0 Dublin, Ireland; johncaird@beaumont.ie

**Keywords:** invasive aspergillosis, CNS aspergillosis, (1,3)-β-d-glucan, isavuconazole, fungal CNS infection

## Abstract

This case highlights the use of (1,3)-beta-d glucan to direct treatment of a cervical spinal cord *Aspergillus fumigatus* infection in a 22-year-old woman immunocompromised due to steroid and anti-TNF therapy in the context of ulcerative colitis and interferon gamma deficiency. A 4-year treatment course requiring neurosurgical intervention on four occasions and prolonged antifungal therapy, including isavuconazole, resulted in clinical cure with a corresponding decrease in CSF beta-d-glucan to <30 pg/mL. Serum and CSF galactomannan levels were not elevated at any point during the clinical course.

## 1. Introduction

Invasive aspergillosis (IA) infections of the central nervous system (CNS) are most common in immunocompromised patients and can present as endophthalmitis, meningitis, ventriculitis, encephalitis, and brain abscess. *Aspergillus fumigatus* is the offending agent in over 70% of cases, but other subspecies, including *A. niger*, *A. flavus*, and *A. terreus*, have been described [1]. Infections can occur in three main ways: firstly, through direct extension from adjacent sites including ethmoid sinuses, mastoid air spaces, and vertebral and epidural abscess. Secondly, hematogenous seeding from a primary site including lung abscess, and thirdly, direct inoculation through trauma, or iatrogenically from invasive procedures or indwelling devices such as ventriculoperitoneal (VP) shunts [2,3]. As a subgroup, spinal cord aspergillus infections are exceedingly rare. They are generally caused by direct extension of locally invasive spinal or lung-invasive aspergillosis but can also be associated with foci in the cerebrum or cerebellum, presumably from hematogenous spread [4,5]. Rare cases of iatrogenic spinal infections associated with perinatal spinal epidural procedures and contaminated steroid products have been described [6,7,8].

Complications of CNS *Aspergillus* infection include abscess formation, thrombosis, and hemorrhage. Prognosis is generally dismal. In the past, mortality of 86–100% has been reported [9,10]. Outcomes have improved with CNS penetrating anti-*Aspergillus* azoles, and survival was shown to be 31% in a landmark paper describing therapy with a combination of voriconazole and neurosurgery [11]. In a recent report from India, eight out of nine patients treated for CNS aspergillosis with voriconazole survived at 14-month follow-up [12]. Treatment courses can be protracted, lasting months, and are a significant burden on patients due to adverse drug effects, instrumentation, and need for close follow-up [13]. Furthermore, treatment is challenging as reliable laboratory markers that reflect treatment response to CNS infection in real time are lacking. Patient improvement in these cases can be reliant on changes in clinical features, such as neurological signs and neuroimaging, which may not always correlate with active fungal disease. The result may be increased bed days, unnecessarily long courses of antifungal therapy, and an overall increased cost of care and risk to patients. Serum (1,3)-β-d-glucan (BDG) has been shown to be sensitive for non-CNS *Aspergillus* infections and accurately reflects response to treatment [14]. The test is not FDA approved for its use in cerebrospinal fluid (CSF) for CNS infections. Although data are lacking, there is a growing body of evidence to suggest this marker is sensitive for CNS aspergillosis [15,16,17]. This case highlights its potential use in this setting.

## 2. Case

In August 2017, a 22-year-old woman was admitted with a seemingly unprovoked reduced level of consciousness, having been found by her mother at home. Magnetic Resonance Imaging (MRI) with venography revealed severe cerebral venous sinus thrombosis of the straight sinus, and emergency thrombectomy and catheter-directed thrombolysis were performed. On day five of admission, a VP shunt was inserted due to obstructive hydrocephalus. With improvement in level of consciousness around day 42, the patient revealed that she had been having low-grade headaches and neck pain in the weeks preceding admission, which were ongoing. There were no other symptoms at the time, and a recent trip to the Greek islands was uneventful. The patient had a background history of ulcerative colitis, requiring multiple courses of corticosteroids in the preceding 6 months and the commencement of adalimumab in the months preceding admission. Other medications included the combined oral contraceptive pill. The patient was having daily isolated fevers >38.5 °C. There were no focal clinical signs on examination at this time apart from limited neck range of motion. A timeline of subsequent events is given in Figure 1.

Initial laboratory findings were as follows: QuantiFERON, EBV serology, CMV serology, and cryptococcal antigen testing were all negative, as was serum galactomannan (GM) index (0.4, range >0.5–1.0). Serum BDG was positive (337 pg/mL, range 30–59 pg/mL). Initial lumbar puncture results showed leukocytes 465 cells/mm^3^ (range 0–5 cells/mm^3^, 73% polymorphs), raised protein 122 mg/dL (CSF range: 15–45 mg/dL), and low glucose 1.6 mmol/L (range 3.5–7.7 mmol/L); no organisms were seen or cultured. A CSF *FILMARRAY^®^ ME* Panel (*Streptococcus pneumoniae*, *Haemophilus influenza*, *Neisseria meningitidis*, *Escherichia coli* K1, *Streptococcus agalactiae*, *Listeria monocytogenes*, *HSV1*, *HSV2*, *VZV*, *HHV-6*, *Enteroviruses*, *CMV*, *Paraechovirus*, *Cryptococcus neoformans/gatii*) was negative. Other CSF tests, including *West Nile Virus*, *JC virus*, GM (0.09, range >0.5), *Aspergillus* PCR, *Candida* PCR, 16 s ribosome, 18 s ribosome, AFB, *GeneXpert*, TB cultures, and *Toxoplasma* PCR were also negative. CSF BDG was >500 pg/mL.

An MRI brain scan was performed in the context of the neck pain, which showed hyperintensity of the cord at C1–C2 level, Figure 2b(L). Initial treatment included empiric antimicrobial therapy with ceftriaxone, vancomycin, and liposomal amphotericin B (Ambisome^®^), which was subsequently switched to voriconazole due to infusion reaction. With the development of weakness of the left upper limb on day 97, repeat imaging showed significant enlargement of the cervical mass and emergency C2–C4 laminectomy and decompressive surgery was performed. Histopathology displayed chronic inflammation only and no other diagnostics for TB or bacterial or fungal infection yielded a diagnosis. Ambisome^®^ was reintroduced at this time in place of voriconazole. Unfortunately, the patient experienced further weakness in both upper limbs, ataxia, and upper motor signs of her right lower limb on day 194 and required further C1 laminectomy and mass excision. At this time, the patient was on amikacin, vancomycin, and Ambisome^®^, in addition to empiric TB treatment. Necrotizing granulomas, septate hyphae, and *A. fumigatus* PCR positivity were findings from the histopathological samples and therapy was rationalized to voriconazole monotherapy.

Initially, the patient improved clinically and radiologically. The development of hypertension attributed to voriconazole, despite normal trough levels 1.65 mcg/mL (range 1.0–5.5 mcg/mL), prompted a change in therapy in July 2018 (day 332) to Ambisome^®^ (intravenously via the outpatient antimicrobial therapy (OPAT) service) and oral isavuconazole. Follow-up CSF testing at that time showed raised WCC 253 cells/mm^3^ (range 0–5 cells/mm^3^, 66% polymorphs), protein 3331 mg/dL (CSF range 35–45 mg/dL), and BDG 103 pg/mL. CSF GM was negative. An UC flare in January 2019 prompted the use of steroids for over 1 month. Cognizance of the ongoing risk of immunosuppression in the context of invasive fungal CNS infection resulted in the decision to perform total colectomy in February 2019. The patient was admitted again in April 2019 with new neurology in the form of leg weakness and loss of anal tone. MRI findings suggestive of progression of infection from skull base to T11, new leptomeningeal nodular enhancement, and a rise in CSF BDG > 500 pg/mL prompted the addition of high-dose steroids and an increase in Ambisome from 5 to 10 mg/kg dose. CSF GM was not performed at this time, but serum CSF was negative. Short-interval MRI at 9 days showed further progression and intrathecal (IT) amphotericin B was commenced, guided by IDSA coccidioidomycosis protocol [18]. An Ommaya device was inserted to facilitate regular administration of intrathecal amphotericin B. These subcutaneous implantable devices provide a secure route of drug delivery into the CSF via an intraventricular catheter allowing for ease of administration and minimization of patient discomfort from repeated lumbar puncture. Like with any non-native material, there is a risk of associated infection, especially related to recent reservoir access. At this time, this patient was found to have very low production of IFN gamma, in particular in response to the fungal cell surface components zymosan and beta-glucan. The IFN gamma response to beta-glucan was 28.8 pg/mL when the control response was 670.8 pg/mL. Recombinant interferon gamma-1b was given by subcutaneous administration 90 micrograms three times weekly beginning in May 2019 and ending in January 2020.

In July 2019, interval imaging showed abnormality confined to the cervicothoracic cord and adjacent dura. The patient was discharged with a plan for self-administered OPAT of Ambisome^®^ 3 mg/kg daily with attendance to hospital three times weekly for IT amphotericin B with a staged increase in IT amphotericin B up to 1.5 mg over a 15-week period. Worsening left hand strength and paresthesia in the C7 dermatome prompted an MRI in September 2019, which revealed progression of the dural mass with impingement of C7 nerve root. CSF analysis showed WCC <1 cells/mm^3^ (range 0–5 cell/mm^3^), protein >600 mg/dL (range 35–45 mg/dL), glucose of 4.2 mmol/L (range 3.5–7.7 mmol/L), and BDG 202 pg/mL. CSF and serum GM testing were not performed at this time. A C6 laminectomy and mass excision were performed. Necrotizing granulomas were seen histologically, but no diagnostics confirmed *Aspergillus* infection. New intracordal enhancement in October 2019 was found on interval MRI, voriconazole was added to peripheral Ambisome^®^ and IT amphotericin B. Immune reconstitution inflammatory syndrome (IRIS) was also considered at this time as a possible cause of new cord changes, but CSF BDG (171 pg/mL) was still raised at this time with paired normal serum BDG (51 pg/mL). Both serum and CSF GM were negative. Further anterior cervical discectomy was performed to mitigate cord-stretch myelopathy in January 2020. Derangement in liver blood tests prompted a switch from voriconazole to isavuconazole. In addition, renal impairment occurred at this time, which required regular IV fluids and sodium bicarbonate, and a reduction in IV Ambisome^®^ to twice weekly. Isavuconazole and Ambisome^®^ were continued in the outpatient setting. Over the following months, the patient had extensive rehabilitation but required the use of a wheelchair. CSF BDG performed in August 2020 was negative (<30 pg/mL). Interval MRI in September 2020 showed persistent but stable abnormality from C1–4 and adjacent dura. Similar findings were seen on further MRI in April 2021; the patient remained on twice-weekly Ambisome^®^ and isavuconazole throughout this period. By June 2021 CSF WCC was <5 cells/mm^3^, *Aspergillus* PCR was negative, and CSF GM and BDG (<30 pg/mL) were within normal range. Antifungals were discontinued at this time. The patient is currently undergoing rehabilitation with close follow-up.

## 3. Discussion

This case highlights the convoluted clinical course that can be seen with complex CNS *Aspergillus* infections. Treatment in these patients can be complicated due to several factors: underlying immunosuppression or the ongoing need for immunosuppressants (e.g., UC); interactions of antifungals with co-medications, particularly warfarin, rifampicin, and antiepileptics; adverse drug events; acquisition of secondary nosocomial infections; and importantly, the assessment of antifungal therapy effectiveness in real time. In this patient’s treatment course, there were multiple episodes of neurological deterioration with corresponding worsening MRI findings. Discerning whether these changes in clinical status represent worsening fungal infection, post-operative inflammation, IRIS, or secondary infection can be difficult and can result in additional antimicrobials, corticosteroids, lumbar punctures, and even neurosurgical intervention. At pivotal times of clinical deterioration in this case, raised CSF BDG pointed towards active fungal infection and directed therapeutics towards escalation of antifungals and appropriate debulking surgery. As serum and CSF GM and serum BDG tests were persistently negative throughout this patient’s course, CSF BDG was the only marker for active fungal disease. Furthermore, a persistently normal BDG for over a 10-month period in the context of a clinically stable course despite persistent change on neuroimaging was supportive of cessation of antifungal therapy.

The use of BDG, a fungal cell wall polysaccharide, in CSF analysis as a reliable marker of fungal CNS infection is gaining traction [15]. High sensitivities have been reported for some fungal infections; coccidioidal meningitis (96%), iatrogenic spinal steroid injection *Exserohilum rostratum* infection (>95%), and cryptococcal meningitis (89%) [15,19]. One study of 92 participants (including 66 controls that did not have fungal infection) showed controls had two-fold lower CSF than serum BDG, whereas the ratio of CSF BDG relative to serum BDG is increased by a factor of 25 in fungal CNS infections [20]. CSF BDG has been used to closely track response to therapy of *Aspergillus* meningitis and ventriculitis in case reports and appears to have greater dynamic analytical range than CSF GM for this purpose [16,17]. In one such case, CSF GM normalized within 2 months despite persistence of symptoms; CSF BDG was <30 pg/mL at 24 months and clinically correlated with cure; treatment was stopped at that time [17]. In general, GM is raised in cases of CNS meningitis, a case series of 93 patients with *Aspergillus* meningitis or encephalitis showed a median index of 6.58 (range 0–1.0) and CSF levels were always higher than serum [21]. GM does have the advantage of specificity for aspergillus infections. BDG is a marker for other fungal infections, including *Coccidoides* spp., *Candida* spp., *Cryptococcus* spp., *Pneumocystis*, and *Fusarium*. Conversely, this case also highlights that despite severe infection with abscess formation, dural involvement and the need for multiple debulking procedures, GM was never raised in CSF or serum. In this setting of negative GM testing, use of CSF BDG can be useful to guide treatment progress and has been described in another case of otitis with meningeal extension [22].

This case also supports the use of isavuconazole for fungal CNS infections. Although isavuconazole is non-inferior to voriconazole for non-CNS IA, data for CNS infection are lacking [23]. Although no exposure thresholds and no exposure-toxicity levels have been established in CSF or serum, isavuconazole does reach therapeutic levels in CSF [24]. CSF isavuconazole levels were checked in this case at a time when infection had spread extensively along the spinal cord and there was concern for potential inadequate penetration of isavuconazole. Evidence of a non-negligible level of isavuconazole (1.46 mg/L) in the CSF in this case was reassuring. A retrospective analysis of 36 patients, mostly from the VITAL or SECURE clinical trials, showed efficacy for a number of fungal CNS infections including IA (*n* = 8) [25]. Additionally, in a case series of nine patients with coccidioidal CNS infection, all required a switch of therapy to isavuconazole from voriconazole due to treatment failure (*n* = 1) or adverse effects (*n* = 8) and had improvement or stabilization of disease between 138 and 810 days of therapy [26]. Two cases of CNS aspergillosis (frontal, frontoparietal) have been successfully treated with isavuconazole [27,28]. As far as the authors are aware, this is the first case of treatment of an *Aspergillus* medullary spinal abscess using isavuconazole in conjunction with low-dose Ambisome^®^. The reduced adverse effect profile and the need for prolonged treatment courses in CNS infections may see this triazole preferentially used if efficacy can be shown in future studies. This case also highlights the role of CSF BDG in diagnosing and managing such a complex disease.

## Figures and Tables

**Figure 1 jof-08-00557-f001:**
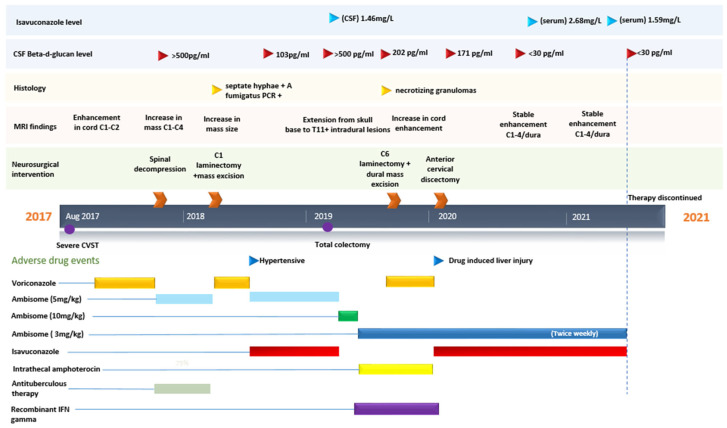
Timeline of major events and drug therapy.

**Figure 2 jof-08-00557-f002:**
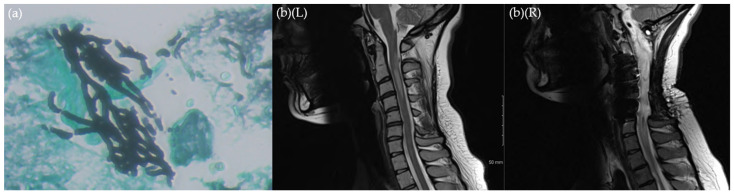
(**a**) Grocott stain of excised spinal cord mass in February 2018 showing septate branching hyphae, also PCR + for *A. fumigatus*; (**b**) sagittal C-spine MRI findings at baseline August 2017 (**L**) and at end of treatment June 2021 (**R**).

## Data Availability

Not applicable.

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
