# Peer review of "Isavuconazole Treatment of Spinal Cord Invasive Aspergillosis Guided by Cerebrospinal Fluid (1,3)-β-d-Glucan Levels in a Patient with Low Interferon-Gamma and Ulcerative Colitis"

_jof, 2022, doi:10.3390/jof8060557_

Round 1

Reviewer 1 Report

Thank you very much for this article.

This is a case of interest. However, it would be useful to add the data requested in the text.
It would also be useful to know the brand of kits used; some differences have been reported between those available from 1-3BG.

kind regards.

Author Response

Response to Reviewer 1

  1. ‘It would be useful to add the data requested in the text.’ (Galactomannan results)

Thank you very much for this suggestion.

Available Aspergillus galactomannan results have now been included in the text at the highlighted times. (Pg 3, line 110, line 117, line 140 and line 147). All results were negative.

  1. ‘It would also be useful to know the brand of kits used; some differences have been reported between those available from 1-3BG’

Thank you for this advice. The Associates of Cape Cod Fungitell ® assay was used. This is an in vitro diagnostic test for the qualitative detection of (1–3)-beta-D-Glucan (BDG) in serum. This information has now been included in the manuscript. (Pg 2, line 59)

Reviewer 2 Report

Complicated case report of an immunocompromised host with spinal aspergillosis successfully managed with multiple surgeries, serial imaging, CSF beta D glucan monitoring and prolonged antifungal therapy with multiple drugs over several years. A key message- CSF BDG can be a useful monitoring tool in CNS aspergillosis.

Queries-

1-Is the CSF BDG measurement standardized and reproducible? If so, pl provide reference if available.

2-Title- includes Low interferon gamma. ?Value/relevance. Pt was given recomb gamma interferon. Duration/dose are not depicted in Figure. If relevant, pl include, otherwise ?omit in text and title.

3-Isavuconazole levels in serum and csf- add comment.

4-Is there a need for chronic antifungal suppression in this compromised host? Concern for relapse with ongoing immunosuppressive therapy.

Author Response

Response to Reviewer 2

  1. ‘Is the CSF BDG measurement standardized and reproducible? If so, pl provide reference if available.’

Thank you for this comment. The CSF BDG measurement is not standardized and reproducible. The Fungitell® test is not FDA approved for its use in CSF for CNS fungal infections. Although data are lacking, there is a growing body of evidence to suggest (1,3)-beta-d-glucan is sensitive for CNS Aspergillosis. This has been described in the introduction section of the original manuscript. (Pg 2, line 57)

  1. ‘Title- includes Low interferon gamma. ?Value/relevance. Pt was given recomb gamma interferon. Duration/dose are not depicted in Figure. If relevant, pl include, otherwise ?omit in text and title.’

Thank you very much for this suggestion. There is no specific value or reference range for low interferon gamma. The interferon gamma response is tested in response to a range of stimuli and this patient was found to have very low production of IFN gamma in particular in response to LPS, IL18 and to the fungal cell surface components zymosan and beta-glucan.  The IFN gamma response to beta-glucan/IL12 was 28.8 pg/ml when the control response was 670.8 pg/ml. Recombinant interferon gamma-1b was given by subcutaneous administration 90 micrograms three times weekly beginning in May 2019 and ending in January 2020. The text and associated timeline image (Figure 1) have been updated to include this recommendation. (Pg 3, line 126)

  1. Isavuconazole - not recommended to have Therap Drug Monitoring with this drug. No optimal levels have been recommended either in serum or other specimens like the CSF. Hence a comment is appropriate regarding utility, reference and interpretation etc.’

Thank you very much for highlighting this. Although therapeutic drug monitoring is not routinely recommended, evidence of isavuconazole treatment of spinal cord invasive Aspergillosis has not been previously described. This patient’s spinal anatomy was abnormal due to infection and previous surgeries. At a time when infection had spread extensively along the spinal cord, we were concerned for potential inadequate penetration of isavuconazole. Although no exposure-thresholds and no exposure-toxicity levels have been established, we were reassured in this case by evidence of a non-negligible level of isavuconazole(1.46mg/L) in the CSF. On your advice, this decision has now been described in more detail (Pg 4, line 202)

‘Is there a need for chronic antifungal suppression in this compromised host? Concern for relapse with ongoing immunosuppressive therapy.’

Thank you for this comment. Although we did not elaborate on this part of the patient’s journey, this issue has been described briefly in the original manuscript (pg 3, line 105). During an ulcerative colitis flare in 2019, a decision was made by the patient, gastroenterology, infectious diseases and colorectal surgery teams to perform total colectomy in order to manage symptoms and remove further need for immunosuppression to support treatment of the Aspergillosis infection. No further symptoms have been experienced since this time and immunosuppressive therapy has not been needed. No updates have been made to the manuscript.
